# BMAD: BENCHMARKS FOR MEDICAL ANOMALY DETECTION

## ABSTRACT

Anomaly detection (AD) is a fundamental research problem in machine learning and computer vision, with practical applications in industrial inspection, video surveillance, and medical diagnosis. In medical imaging, AD is especially vital for identifying anomalies that may indicate rare diseases or conditions. Despite its significance, there is a lack of a universal and fair benchmark for evaluating AD methods on medical images, which hinders the development of more generalized and robust AD methods in this specific domain. To bridge this gap, we introduce a comprehensive evaluation benchmark for assessing AD methods on medical images. This benchmark encompasses six reorganized datasets from five medical domains (i.e. brain MRI, liver CT, retinal OCT, chest X-ray, and digital histopathology) and three key evaluation metrics, and includes a total of fifteen state-of-the-art AD algorithms. This standardized and well-curated medical benchmark with the well-structured codebase enables comprehensive comparisons among recently proposed anomaly detection methods. It will facilitate the community to conduct a fair comparison and advance the field of AD on medical imaging.

## 1 INTRODUCTION

Anomaly detection is a technique to identify patterns or instances that deviate significantly from the normal distribution or expected behavior. It plays a crucial role in various real-world applications, including but not limited to, video surveillance, manufacturing inspection, rare disease detection and diagnosis, and autonomous driving, etc. Recent studies in anomaly detection often follows the unsupervised paradigm, where model training relies solely on the availability of normal samples. The absence of abnormal samples within this one-class modeling framework makes the discriminative feature learning challenging. In computational medical image analysis, unsupervised anomaly detection is essential for identifying unusual and atypical anomalies. Here, anomaly can refer to abnormal structures, lesions, or patterns in medical images that may indicate the presence of diseases, tumors, or other medical conditions. In biomedicine, normalities are usually well defined and collecting normal data is comparatively easier; By contrast, anomalies are heterogeneous and it is implausible to gather a comprehensive set of training samples that covers all possible abnormal cases, especially concerning rare diseases and unprecedented anomalies or diseases. This inherent open-set nature of medical data collection implies that conventional supervised methods could result in poor performance on unseen abnormalities. To supplement the above mentioned limitation of supervised learning, the primary objective of medical AD is not to classify known diseases, but rather to signal an alert when abnormalities occur.

Due to the practical significance of anomaly detection, several benchmarks have been established recently (Xie et al., 2023; Zheng et al., 2022; Han et al., 2022; Yang et al., 2022). However, these benchmarks primarily focus on industrial images such as those in MVTec (Bergmann et al., 2019) and natural images, and there is a lack of benchmark datasets specifically designed for the medical field despite its significannce. Among the rich literature of medical anomaly detection, due to the lack of dedicated medical anomaly detection datasets, prior arts usually employ datasets that are initially developed for supervised classification (Schlegl et al., 2017; Zhou et al., 2020a; Zhang et al., 2020; Wang et al., 2017) or segmentation tasks (Chen & Konukoglu, 2018; Baur et al., 2019). For the anomaly detection purpose, these datasets undergo extensive cleaning and reorganization. For one thing, we observe an inconsistency in the citation of data sources (Salehi et al., 2021; Zhou et al., 2020b; Chen et al., 2022). For another, even using the same dataset, there is no clear consensus

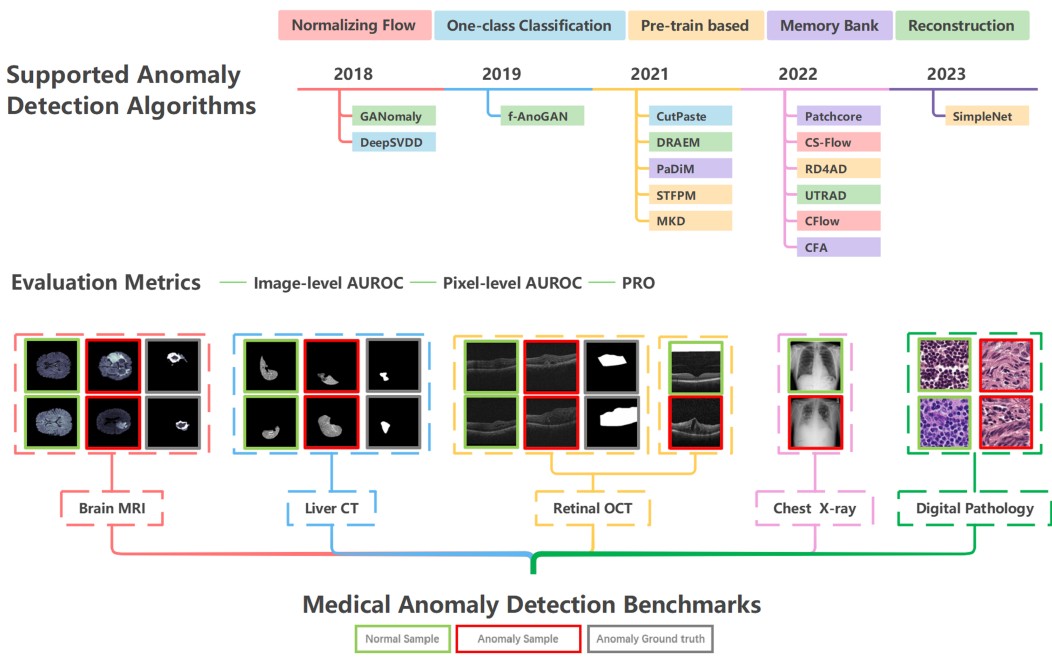

Figure 1: Diagram of the BMAD benchmarks. BMAD includes six datasets from five different domains for medical anomaly detection, among which three support pixel-level AD evaluation and the other three for sample-level assessment only. BMAD provides a well-structured and easy-used code base, integrating fifteen SOTA anomaly detection algorithms and three evaluation metrics.

or explanation on how to reorganize the dataset so that it can be usable for anomaly detection and localization (Pinaya et al., 2021; Rai et al., 2021; Wolleb et al., 2022). As a result, a fair comparing among these methods is difficult. Therefore, it is crucial to have standardized datasets and evaluation metrics specifically tailored to this field. These resources facilitate comprehensive assessments and the advancement of anomaly detection techniques for medical applications.

To address the aforementioned issues, we introduce a uniform and comprehensive evaluation benchmark, namely BMAD [1], for assessing anomaly detection methods on medical images. This benchmark encompasses six well-reorganized datasets from five medical domains (i.e. brain MRI, liver CT, retinal OCT, chest X-ray, and digital histopathology) and three key evaluation metrics, and includes a total of fifteen state-of-the-art (SOTA) AD algorithms. This standardized and well-curated medical benchmark with the well-structured codebase enables comprehensive comparisons among recently proposed anomaly detection methods. Afterward, we evaluate the fifteen SOTA algorithms over the benchmarks and provides in-depth discussions on the results, which pinpoints potential research directions in future. Our contributions in this work can be summarize as following:

- We constructed a unified and standardized benchmark that encompasses six benchmarks from five common medical domains. Great efforts were taken to reorganize and adapt the datasets to the unsupervised anomaly detection setting in computational medical imaging.

- We built a well-structure, easy-use codebase supporting 15 SOTA anomaly detection algorithms and their evaluations.

- We comprehensively analyzed algorithms' cons and pros on BMAD datasets. The observations and discussions would encourage researchers to develop better AD models for medical data.

---

[1] A Creative Commons Attribution-NonCommercial-ShareAlike (CC BY-NC-SA) license is issued to BMAD. This license complies with all the original dataset's licenses.

## 2 RELATED WORK

For unsupervised anomaly detection, the existing algorithms can be categorized into two paradigms: data reconstruction-based approaches and feature embedding-based (or projection-based) approaches. The former typically compares the differences between the reconstructed data and the original data in the data space to identify potential anomalies, while the latter infers anomalies by analyzing the abstract representations in the embedding space.

### 2.1 RECONSTRUCTION-BASED METHODS

A reconstruction-based approach usually deploys a generative model for data reconstruction. It targets for small reconstruction residues for normal data, but large errors for anomalies. The distinction in reconstruction errors forms the basis for anomaly detection. **AutoEncoder (AE)** and **Variational AE** have been the first and most popular models for this purpose (Sakurada & Yairi, 2014; Zhou & Paffenroth, 2017; Bergmann et al., 2018; Sabokrou et al., 2018; Li et al., 2019; Gong et al., 2019; Kascenas et al., 2022; Park et al., 2020; Hou et al., 2021; Zhou et al., 2021). Later, **Generative Adversarial Networks (GANs)** are used to replace AE for its high-quality output (Sabokrou et al., 2018; Perera et al., 2019; Schlegl et al., 2017; Akcay et al., 2018; Schlegl et al., 2019; Yan et al., 2021). Recently, there is a trend of exploiting **diffusion models** for normal sample generation (Wyatt et al., 2022; Teng et al., 2022; Wolleb et al., 2022). In addition to convolutional neural networks, the transformer architecture is also explore in latest studies to build these generative models (You et al., 2023; Jin et al., 2023; Lee & Kang, 2022; Chen et al., 2022).

To improve AD performance, regularization strategies are incorporated into normal sample reconstruction. Following the idea of denoising AE, Gaussian noise is added into normal samples for a better normal data restoration performance (Cao et al., 2016a;b; Kascenas et al., 2022). In the masking mechanism, a normal sample is randomly masked and then inpainted back (Li et al., 2020; Zavrtanik et al., 2021b; Yan et al., 2021). Furthermore, many studies focus on synthesizing abnormalities on normal training samples and use the generative model to restore the original normal version (Zavrtanik et al., 2021a; Deng & Li, 2022a; Tan et al., 2022). Recently, the memory mechanism is exploited to further constrain model's capability on reconstructing abnormal samples (Gong et al., 2019; Park et al., 2020; Hou et al., 2021; Zhou et al., 2021).

### 2.2 PROJECTION-BASED METHODS

A projection-based method employs either a task-specific model or simply a pre-trained network to map data into abstract representations in an embedding space, enhancing the distinguishability between normal samples and anomalies.

**One-class classification** usually uses normal support vectors/samples to define a compact closed one-class distribution. The one-class SVM (Schölkopf et al., 2001) sought a kernel function to map the training data onto a hyperplane in the high-dimensional feature space. Any samples not on this hyperplane are considered anomalous samples. Similarly, support vector data description (Tax & Duin, 2004), DeepSVDD (Ruff et al., 2018), and PatchSVDD (Yi & Yoon, 2020) aimed to find a hyper-sphere enclosing normal data using either kernel-based methods or self-supervised learning.

**Teacher-student (T-S)** architecture for knowledge distillation is a prevalent approach in AD recently (Bergmann et al., 2020; Cao et al., 2022; Rudolph et al., 2023; Salehi et al., 2021; Yamada & Hotta, 2021; Deng & Li, 2022b; Tien et al., 2023). Since the student network only learn representations of normal samples from the teacher, it may not follow the teacher's behavior for abnormal cases. The representation discrepancy of the T-S pair forms the basis of anomaly detection.

**Memory Bank** is a mechanism of remembering numerical prototypes of the training date (Li et al., 2021b; Defard et al., 2021; Roth et al., 2022; Lee et al., 2022). Then various algorithms such as KNN or statistical modeling are used to determine the labels for queries.

**Normalizing Flow** is a method to explicitly model data distribution (Rezende & Mohamed, 2015). For AD, a flow model maps normal features onto a complex invertible distribution. In inference, normal samples are naturally localized into the trained distribution range, while abnormal samples are projected onto a separate distribution range (Yu et al., 2021; Rudolph et al., 2022; 2021; Gudovskiy et al., 2022).

| Benchmarks | Originations | Total | Train | Test | Validation | Sample size | Annotation Level |
|---|---|---|---|---|---|---|---|
| Brain MRI | BraTS2021 (Baid et al., 2021) | 11,298 slices | 7,500 | 3,715 | 83 | 240*240 | Segmentation mask |
| Liver CT | BTCV (Landman et al., 2015) + LiTs (Bilic et al., 2019) | 3,201 slices | 1,542 | 1,493 | 166 | 512*512 | Segmentation mask |
| Retinal OCT | RESC (Hu et al., 2019) | 6,217 images | 4,297 | 1,805 | 115 | 512*1,024 | Segmentation mask |
| | OCT2017 (Kermany et al., 2018) | 27,315 images | 26,315 | 968 | 32 | 512*496 | Image label |
| Chest X-ray | RSNA (Wang et al., 2017) | 26,684 images | 8,000 | 17,194 | 1,490 | 1,024*1,024 | Image label |
| Pathology | Camelyon16 (Bejnordi et al., 2017) | 7,321 patches | 5,088 | 1,997 | 236 | 256*256 | Image label |

Table 1: Summary of the six benchmarks from five imaging domains in BMAD.

## 3 BENCHMARKS, METRICS, AND ALGORITHMS

### 3.1 MEDICAL AD BENCHMARKS

When constructing this benchmark, we had following considerations in dataset selection: diversity of imaging modalities, diversity of source domains/organs, and license for data reorganization, remix and redistribution. Specifically, our BMAD includes six medical benchmarks from five different domains for medical anomaly detection. We summarize these benchmarks in Table 1. Within these benchmarks, three supports pixel-level evaluation of anomaly detection, while the remaining three is for sample-level assessment only. More details about the original benchmark datasets, their licenses, and our efforts on data reorganization are provided in the Appendix A.

**Brain MRI AD Benchmark.** Magnetic Resonance Imaging (MRI) imaging is widely utilized in brain tumor examination. The Brain MRI AD benchmark is reorganized using the flair modality of the latest large-scale brain lesion segmentation dataset, BraTS2021 (Baid et al., 2021). BraTS2021 is proposed for multimodal brain tumor segmentation. The original data comprises a collection of the complete 3D volume of a patient's brain structure and corresponding brain tumor segmentation annotation. To adapt the data to AD, we sliced both the brain scan and their annotation along the axial plane. Only slides containing substantial brain structures, usually with a depth of 60-100, were selected in this benchmark. Slices without brain tumor are labelled as normal. Then normal slices from a subset of patients form the training set, and the rest are divided into validation and test sets. To avoid data leakage in model evaluation, we leveraged the information of patient IDs for data partition and ensured that data from the same patient was contained by one set only.

**Liver CT AD Benchmark.** Computed Tomography (CT) is commonly used for abdominal examination. We structure this benchmark from two distinct datasets, BTCV (Landman et al., 2015) and Liver Tumor Segmentation (LiTs) set (Bilic et al., 2019). The anomaly-free BTCV set is initially proposed for multi-organ segmentation on abdominal CTs and taken to constitute the train set in this benchmark. CT scans in LiTs is exploited to form the evaluation and test data. For both datasets, Hounsfield-Unit (HU) of the 3D scans were transformed into grayscale with an abdominal window. The scans were then cropped into 2D axial slices, and the liver regions were extracted based on the provided segmentation annotations. Following conversion in prior arts (Dey, 2021; Li et al., 2022), we further performed histogram equalization on each slide for image enhancement [2]

**Retinal OCT AD Benchmarks.** Optical Coherence Tomography (OCT) is commonly used for scanning ocular lesions in eye pathology. To cover a wide range of anomalies and evaluate anomaly localization, we reorganize two benchmarks, Retinal Edema Segmentation Challenge dataset (RESC) (Hu et al., 2019) and Retinal-OCT dataset (OCT2017) (Kermany et al., 2018). RESC, originally published to segment retinal edema in retinal OCT, is reorganized for benchmarking anomaly localization. The original training, validation, and test sets contain 70, 15, and 15 cases, respectively. Each case includes 128 slices, some of which suffer from retina edema. We utilized the provided segmentation annotation to identify normal and abnormal samples. To avoid data leakage, slices from the same subject only appear in either validation or test set. OCT2017 is a large-scale classification dataset originally. Images are categorized into 4 classes: normal, Choroidal Neovascularization, Diabetic Macular Edema, and Drusen Deposits. To construct this sample-level

---

[2]For completeness, we also provide a version of the liver benchmark without any data processing in BMAD. This allow researchers to access both versions and make informed decisions based on their specific needs. For reference, quantitative evaluation of the 15 AD algorithms on this dataset is provided in the Appendix D.

anomaly detection benchmark, we use the disease-free samples in the original OCT2017 training set as our training data. For images in the original test set, images in the 3 diseased classes are labeled as abnormal. Stratified sampling is adopted to form the evaluation and test sets.

**Chest X-ray AD Benchmark.** X-ray imaging is widely used for examining the chest and provides precise thoracic data. This original chest CT dataset is created for levering ML models for chest X-Ray diagnosis (Wang et al., 2017). The lung images are associated with nine labels: Normal, Atelectasis, Cardiomegaly, Effusion, Infiltration, Mass, Nodule, Pneumonia and Pneumothorax, which covers the eight common thoracic diseases observed in chest X-rays. To reorganize the dataset for anomaly detection, we labelled images in the abnormal categories as abnormal. We follow the original datasheet and split the data into train, test, and validation sets for anomaly detection.

**Histopathology AD Benchmark.** Histopathology involves the microscopic examination of tissue samples to study and diagnose diseases such as cancer. We utilize Camelyon16 (Bejnordi et al., 2017), a digital pathology imaging breast cancer metastasis detection dataset, to build the histopathology benchmark. Camelyon16 contains 400 40x whole-slide images (WSIs) stained with hematoxylin and eosin, accompanied by multiple versions in a lower magnification. Annotations of metastasis on WSIs are provided. Considering their unique characteristics of WSI such as large size, we follow convention in prior arts (Li & Ping, 2018; Tian et al., 2019; He & Li, 2023) and opted to assess AD models at the patch level in 40X. Specifically, we randomly cropped 5,088 normal patches from the 160 normal WSIs in the original training set of Camelyon16, forming the training samples in the benchmark. For the validation set, we cropped 100 normal and 100 abnormal patches from the 13 validation WSIs. Similarly, for testing, 1k normal and 1k abnormal patches were cropped from the 115 testing WSIs from the original Camelyon16 dataset.

**Ethical and fairness concerns in data.** Among the 7 original datasets used to construct BMAD, RESC was collected in China, the rests from advanced countries including America, Germany, Danmark, etc. This gives rise to inherent geographical and sampling biases, which inevitably exerts some impact on the evaluation outcomes.

## 3.2 Evaluation Metrics

Anomaly detection can be evaluated from the sample level (i.e., detection rate) and the pixel level (i.e., anomaly localization). In BMAD, we take the area under the ROC curve (AUROC) as the major metric to quantify the sample-level and pixel-level performance. It quantifies the trade-off between True Positive Rate (TPR) and False Positive Rate (FPR) across different decision thresholds. For **sample-level AUROC**, anomaly score is calculated based on the specific algorithm design and different thresholds are applied to determine if a sample is normal or abnormal. The obtained TPR and FPR pairs are recorded for estimating the ROC curve and AUROC value. To calculate the **pixel-level AUROC**, different thresholds are applied to the anomaly map. If a pixel has an anomaly score greater than the threshold, the pixel is anomalous. Over an entire image, the corresponding TPR and FPR pairs are used for numerical calculation.

Note that AUROC has limitations to evaluate samll tumor localization, as incorrect localization of smaller defect regions has a minimal impact on the metric. To address this issue, we follow prior arts (Bergmann et al., 2019; Deng & Li, 2022b; Tien et al., 2023) and include another threshold-independent metric, **per-region overlap (PRO)**, for anomaly localization evaluation. PRO treats anomaly regions of different size equally, up-weighting the influence of small-size abnormality localization in evaluation. Specifically, for each threshold, detected anomalous pixels are grouped into connected components and then PRO averages localization accuracy over all components.

**Remark:** DICE coefficient sometimes is explored in medical anomaly detection. However, as we stated in the Appendix D, DICE coefficient is a threshold-dependent metric. The threshold's optimal value varies depending on algorithms and specific tasks. Therefore, we opted not to include the DICE comparison in the main benchmark. Instead, we reported the DICE values of the 15 algorithms in the Appendix D for reference.

## 3.3 Supported AD Algorithms

BMAD integrates fifteen SOTA anomaly detection algorithms, among which four are reconstruction-based methods and the rest eleven are feature embedding-based approaches. Among

the reconstruction-based methods, **AnoGAN** (Schlegl et al., 2017) and **f-AnoGAN** (Schlegl et al., 2019) exploit the GAN architecture to generate normal samples. **DRAEM** (Zavrtanik et al., 2021a) adopts an encoder-decoder architecture for abnormality inpainting. Then a binary classifier takes the original data and the inpainting result as input for anomaly identification. **UTRAD** (Chen et al., 2022)treated the deep pre-trained features as dispersed word tokens and construct an autoencoder with transformer blocks. Among the projection-based methods, **DeepSVDD** (Ruff et al., 2018), **CutPaste** (Li et al., 2021a) and **SimpleNet** (Liu et al., 2023) are rooted in one-class classification. DeepSVDD searches a smallest hyper-sphere to enclose all normal embeddings extraced from a pre-tarined model. CutPaste and SimpleNet introduce abnormality synthesis algorithms to extend the one-class classification, where generated abnormality synthesis is taken as negative samples in model training. Motivated by the paradigm of knowledge distillation, **MKD** (Salehi et al., 2021) and **STFPM** (Yamada & Hotta, 2021) leverage multi-scale feature discrepancy between the teacher-student pair for AD. Instead of adopting the similar backbones for the T-S pair in knowledge distillation, **RD4AD** (Deng & Li, 2022b) introduced a novel architecture consisting of a teacher encoder and a student decoder, which significantly enlarges the representation dissimilarity for anomaly samples. All of **PaDiM** (Defard et al., 2021), **PatchCore** (Roth et al., 2022) and **CFA** (Lee et al., 2022) rely on a memory bank to store normal prototypes. Specifically, PaDiM utilizes a pre-trained model for feature extraction and models the obtained features using a Gaussion distribution. PatchCore leverages core-set sampling to construct a memory bank and adopts the nearest neighbor search to vote for a normal or abnormal prediction. CFA improves upon PatchCore by creating the memory bank based on the distribution of image features on a hyper-sphere. As notable from the name, **CFlow** (Gudovskiy et al., 2022) and **CS-Flow** (Rudolph et al., 2022) are flow-based methods. The former introduced positional encoding in conjunction with a normalizing flow module and the latter incorporates multi-scale features for distribution estimation.

## 4 Experiments and Discussions

### 4.1 Implementation Details

When evaluating the fifteen AD algorithms over the BMAD benchmarks, we follow their original papers and try their default hyper-parameter settings first. If a model doesn't converge during training and requires hyper-parameter tuning, we try the combination of following common settings, which include 3 learning rate ($10^{-3}$, $10^{-4}$ and $10^{-5}$), 2 optimizer (SGD and Adam), and 3 thresholds for anomaly maps (0.5, 0.6, and 0.7)[3]. For each converged model, we monitor the training progress and record the validation accuracy every 10 epochs. The final evaluation is carried out on the test set using the best checkpoint selected by the validation sets. To visualize anomaly localization results, we employ min-max normalization on the obtained anomaly maps. This ensures the effects of all algorithms appropriately displayed and facilitates the comparison of anomaly localization across different methods. Notably, for a reliable comparison, we repeat the training and evaluation five times, each with a different random seed, and report the mean and standard deviation of the numerical metrics. In this study, all experiments are performed on a workstation with 2 NVIDIA RTX 3090 GPU cards.

### 4.2 Results and Discussions

**Experimental result overview.** The numerical results of anomaly detection and localization over the BMAD benchmark are summarized in Table 2, where the top three performance along each metric are highlighted by underlining. We also provide visualization examples of anomaly localization results in Fig. 2, where redness corresponds to a high anomaly score at the pixel level. Although no single algorithm consistently outperforms others, overall, the feature-based methods shows better performance than the reconstruction-based methods. We believe that two reasons may lead to this observation. First, applying generative models to anomaly detection usually relies on model's reconstruction residue in the pixel level. However, a well-trained generative model usually has good generalizability and it has been found in prior arts that certain anomalous regions can be well reconstructed. This issue hurts anomaly detection performance. Second, reconstruction residue in the pixel level may not well reflect the high-level, context abnormalities. In contrast, algorithms detecting abnormalities from the latent representation domain (such as RD4AD (Deng &

---

[3]Please refer to the Appendix B for the specific hyper-parameter setting for each algorithm.

| Benchmarks | BraTS2021 | | | BTCV + LiTs | | | RESC | | | OCT2017 | RSNA | Camelyon16 |
|---|---|---|---|---|---|---|---|---|---|---|---|---|
| | Image AUROC | Pixel AUROC | Pixel Pro | Image AUROC | Pixel AUROC | Pixel Pro | Image AUROC | Pixel AUROC | Pixel Pro | Image AUROC | Image AUROC | Image AUROC |
| Image Reconstruction-based Methods | | | | | | | | | | | | |
| f-AnoGAN (Schlegl et al., 2019) | 77.26 ± 0.18 | NA | NA | 58.53 ± 0.21 | NA | NA | 77.42 ± 0.85 | NA | NA | 73.42 ± 1.85 | 55.15 ± 0.09 | 69.49 ± 1.98 |
| GANomaly (Akcay et al., 2018) | 74.79 ± 1.93 | NA | NA | 54.60 ± 3.06 | NA | NA | 52.56 ± 3.95 | NA | NA | 70.47 ± 9.98 | 62.90 ± 0.65 | 54.44 ± 2.57 |
| DRAEM (Zavrtanik et al., 2021a) | 62.35 ± 9.03 | 82.29 ± 4.07 | 63.76 ± 4.16 | 69.95 ± 3.86 | 87.45 ± 3.23 | 79.29 ± 5.66 | 83.22 ± 8.21 | 86.79 ± 3.14 | 63.55 ± 4.62 | 88.03 ± 8.36 | 67.70 ± 1.72 | 52.35 ± 0.77 |
| UTRAD (Chen et al., 2022) | 82.92 ± 2.32 | 92.61 ± 0.67 | 72.29 ± 2.12 | 55.81 ± 5.66 | 87.88 ± 1.32 | 71.12 ± 3.46 | 89.39 ± 1.92 | 94.54 ± 1.24 | 77.49 ± 4.30 | 96.78 ± 0.56 | 75.64 ± 1.24 | 69.96 ± 4.64 |
| Image Feature-based methods | | | | | | | | | | | | |
| DeepSVDD (Ruff et al., 2018) | 86.98 ± 0.66 | NA | NA | 53.96 ± 1.84 | NA | NA | 74.17 ± 1.29 | NA | NA | 76.76 ± 1.37 | 64.48 ± 3.17 | 60.98 ± 1.82 |
| CutPaste (Li et al., 2021a) | 78.81 ± 0.67 | NA | NA | 59.33 ± 4.86 | NA | NA | 90.23 ± 0.61 | NA | NA | 96.76 ± 0.62 | 82.61 ± 1.22 | 75.18 ± 0.41 |
| SimpleNet (Liu et al., 2023) | 82.52 ± 3.34 | 94.76 ± 1.04 | 78.38 ± 3.17 | 72.28 ± 2.68 | 97.51 ± 0.56 | 91.07 ± 1.79 | 76.15 ± 7.46 | 77.14 ± 4.76 | 49.07 ± 5.23 | 94.68 ± 2.17 | 69.12 ± 1.27 | 62.38 ± 3.71 |
| MKD (Salehi et al., 2021) | 81.47 ± 0.36 | 89.44 ± 0.24 | 67.59 ± 0.99 | 60.72 ± 1.19 | 96.06 ± 0.27 | 91.08 ± 0.30 | 89.00 ± 0.25 | 86.74 ± 0.65 | 66.17 ± 1.51 | 96.74 ± 0.26 | 82.01 ± 0.12 | 77.54 ± 0.27 |
| RD4AD (Deng & Li, 2022b) | 89.45 ± 0.91 | 96.45 ± 0.17 | 85.86 ± 0.23 | 60.38 ± 1.17 | 96.01 ± 1.19 | 90.29 ± 2.51 | 87.77 ± 0.87 | 96.18 ± 0.15 | 85.62 ± 0.47 | 97.30 ± 0.79 | 67.63 ± 1.11 | 66.81 ± 0.71 |
| STFPM (Yamada & Hotta, 2021) | 83.04 ± 0.67 | 95.62 ± 0.12 | 83.02 ± 0.44 | 61.75 ± 1.58 | 91.18 ± 5.52 | 90.62 ± 6.87 | 84.82 ± 0.50 | 94.68 ± 0.57 | 81.27 ± 1.49 | 96.76 ± 0.23 | 72.93 ± 1.96 | 66.36 ± 1.01 |
| PaDiM (Defard et al., 2021) | 79.02 ± 0.38 | 94.37 ± 1.03 | 76.41 ± 0.84 | 50.78 ± 0.61 | 90.94 ± 0.84 | 76.79 ± 0.41 | 75.87 ± 0.54 | 91.44 ± 0.42 | 71.68 ± 0.81 | 91.75 ± 0.96 | 77.49 ± 1.87 | 67.25 ± 0.32 |
| PatchCore (Roth et al., 2022) | 91.65 ± 0.36 | 96.97 ± 0.04 | 85.68 ± 0.24 | 60.28 ± 0.76 | 96.43 ± 0.19 | 87.75 ± 0.49 | 91.55 ± 0.10 | 96.48 ± 0.10 | 85.84 ± 0.25 | 98.57 ± 0.03 | 76.14 ± 0.67 | 69.34 ± 0.21 |
| CFA (Lee et al., 2022) | 84.38 ± 0.87 | 96.33 ± 0.14 | 83.78 ± 0.51 | 62.00 ± 1.08 | 97.24 ± 0.14 | 92.75 ± 0.21 | 69.90 ± 0.26 | 91.10 ± 0.87 | 69.77 ± 0.41 | 79.47 ± 0.56 | 66.83 ± 0.23 | 65.64 ± 0.10 |
| CFLOW (Gudovskiy et al., 2022) | 74.82 ± 5.32 | 93.76 ± 0.67 | 75.45 ± 3.53 | 50.80 ± 4.47 | 92.41 ± 1.16 | 83.11 ± 1.28 | 74.95 ± 5.81 | 93.78 ± 0.57 | 76.80 ± 1.72 | 85.35 ± 2.11 | 71.53 ± 1.49 | 55.66 ± 1.97 |
| CS-Flow (Rudolph et al., 2022) | 90.91 ± 0.83 | NA | NA | 59.37 ± 0.54 | NA | NA | 87.34 ± 0.58 | NA | NA | 98.47 ± 0.28 | 83.20 ± 0.46 | 68.38 ± 0.42 |

Table 2: AD performance (mean+STD) comparison over the benchmarks in BMAD. The results are obtained from five repetitions of the experiment. NA indicates that a method doesn't support anomaly localization. The top three methods for each metric are underlined.

Li, 2022b), PatchCore (Roth et al., 2022), etc.) facilitate identifying abstract structural anomalies. Therefore, these algorithms perform much better than the generative models. It should be noted that for benchmarks like Liver CT and Brain MRI, where the background consists mostly of black pixels and the distribution of normal and anomalous samples is imbalanced, the numerical results exist bias. Therefore, a high pixel-level AUROC score may indicate that the model correctly classifies the majority of normal pixels, but it does not necessarily reflect the model's ability to detect anomalies accurately. Besides, we have several interesting observations through this research that necessitate careful analysis in order to advance the field of medical anomaly detection. We elaborate our insights and discoveries as follows.

**Anomaly localization analysis.** Since different approaches generate the anomaly map in various ways, either relying on reconstruction error (Deng & Li, 2022b; Wang et al., 2021; Zavrtanik et al., 2021a; Chen et al., 2022), using gradient-based visualization (Salehi et al., 2021; Roth et al., 2022), or measuring feature discrepancy (Defard et al., 2021; Gudovskiy et al., 2022; Lee et al., 2022), they shows distinct advantages and limitations. Generally speaking, both numerical data in Table 1 and the visualization results in Fig. 2 demonstrate that knowledge-distillation methods, especially RD4AD (Deng & Li, 2022b), achieve better localization performance. Although memory bank-based algorithm, Patchcore (Roth et al., 2022), is more convincing at sample-level detection, its abnormality localization is very coarse. Reconstruction-based algorithms, DRAEM (Zavrtanik et al., 2021a) and UTRAD (Chen et al., 2022), shows diverse performance. We hypothesize their distinct capability of anomaly localization is attributed to the different architecture of CNN and transformer. We notice that DRAEM (Zavrtanik et al., 2021a) is particularly sensitive to texture information, often focusing on regions with significant variations in tumor texture. Since such variations may be distributed across all regions in medical imaging, it partially limits the effectiveness of the proposed approach. CFlow (Gudovskiy et al., 2022) shows bad anomaly localization performance and more investigation is needed for its improvement.

**Model efficiency analysis.** For all fifteen algorithms in BMAD, we conduct a comparative efficiency analysis, in terms of sample-level AD accuracy, inference speed and GPU usage. The results are summarized in Fig 3, where the X-axis refers to the inference time per image and Y-axis denotes the performance of the anomaly detection result. The size of the circle denotes the GPU memory consumption during the inference phase. PatchCore (Roth et al., 2022), RD4AD (Deng & Li, 2022b), and CS-FLOW (Rudolph et al., 2022) emerge as the top 3 models across multiple benchmarks in terms of performance. It should be noted that though CS-Flow demonstrates comparable inference time to the other two models, it has lower efficiency to generate pixel-lever anomaly maps.

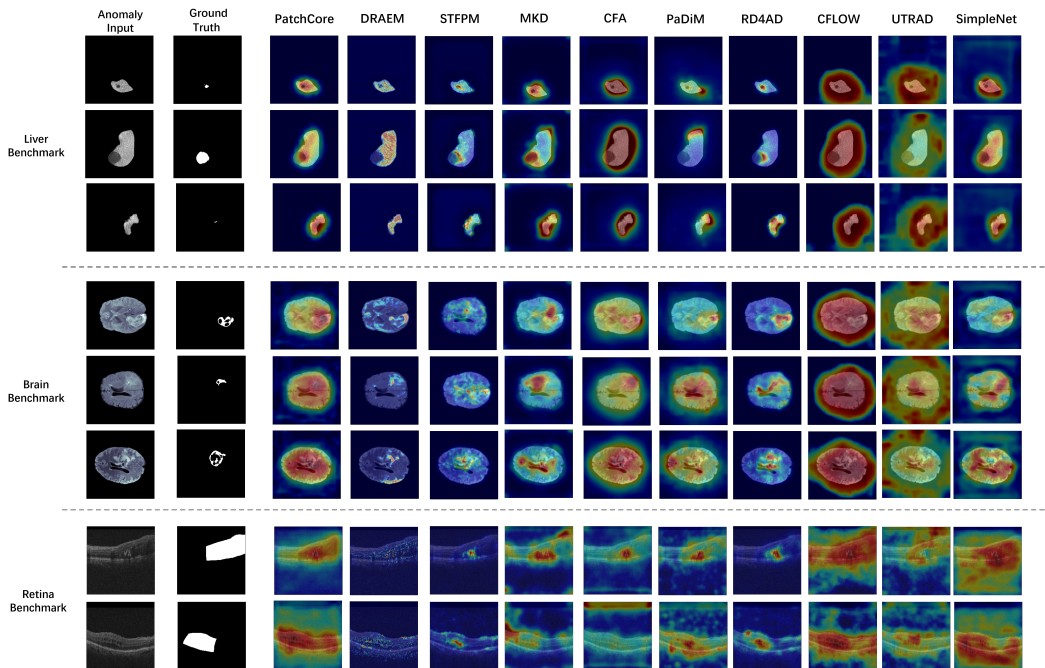

Figure 2: Visualization examples of anomaly localization on the three benchmarks that support pixel-level AD assessment.

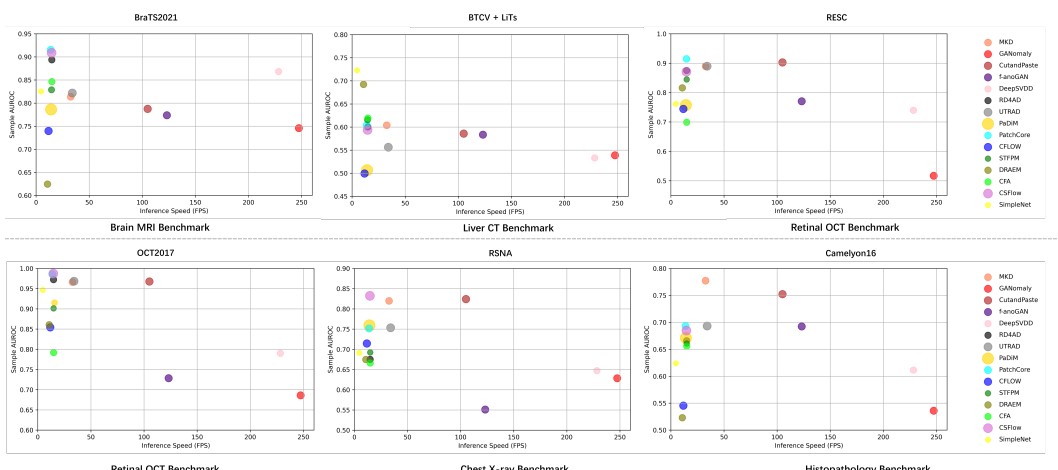

Figure 3: Model Efficiency Analysis. X-axis refers to the average inference time per image and Y-axis denotes anomaly detection accuracy. The size of the circle denotes the GPU memory consumption during the inference phase. In the sub-images, there may be slight variations in the results due to model adjustments like selecting specific parameters and backbones on each benchmark.

**Anomaly synthesis is challenging.** In unsupervised AD methods, one common approach is to synthesize abnormalities to augment model training. CutPaste (Li et al., 2021a) and DRAEM (Zavrtanik et al., 2021) are the examples. However, to address the variability in shape, texture, and color of medical anomalies across different domains, a customized synthesis algorithm is needed to simulate realistic tumor lesions and their distributions. It is important to acknowledge the inherent difficulty in simulating the morphology of anomalies, and this challenge becomes even more pronounced when considering rare diseases. We discovered that the Brain MRI and Liver CT benchmarks are better suited for low-level feature-based anomaly augmentation methods. This observation aligns with the characteristics of the Chest X-ray and Histopathology benchmarks, where abnormalities often ex-

hibit distinct and observable changes in overall structure or appearance. Therefore, it is essential to develop domain-specific approaches that account for these factors when augmenting anomalies in medical image datasets.

**Pre-trained networks significantly contribute to medical domain.** Through there is a continuous debate on if information obtained from natural images is transferable to medical image analysis, our results show that the rich representations of pre-trained models would improve medical anomaly detection by careful algorithm design. Among the models evaluated, algorithms based on the knowledge-distillation paradigm (e.g. MKD (Salehi et al., 2021) and RD4AD (Deng & Li, 2022b)) and memory bank (e.g. Patchcore (Roth et al., 2022)) leverage the powerful feature extraction capabilities of large pre-train models and exhibit better performance in anomaly localization, which plays a crucial role in clinical diagnosis. SimpleNet (Liu et al., 2023) utilized a pre-trained feature extraction module alongside a Gaussian denoising module, proving effective for enhancing low-level feature images. This also suggests that the denoising module operates optimally within the same repersatatation will fit the best for the detection.

**Memory bank-based methods have shown promising performances.** PatchCore (Roth et al., 2022) is a representative example. These methods possess the ability to incorporate new memories, effectively mitigating forgetting when learning new tasks. Hence, the memory bank serves as an ideal rehearsal mechanism. However, these methods have specific hardware requirements to ensure efficient storage and retrieval of stored information. Achieving high-capacity storage systems and efficient memory access mechanisms for optimal performance while minimizing interference time presents a notable challenge. Furthermore, our observations indicate that memory-based methods, while sensitive to global anomalies, may not excel in terms of localizing and visualizing anomalies when compared to feature reconstruction methods. Accurate anomaly localization holds crucial practical value for AD algorithms and provides valuable insights to medical professionals. Therefore, memory bank-based methods may encounter challenges and limitations that impact their competitiveness in certain scenarios.

**Model degradation problem.** The model degradation problem occurs when a deep neural network, trained on a large dataset, exhibits degraded performance as the network's depth increases. We have also found that model degradation issues also exist in BMAD. However, it is a challenge to add appropriate preprocessing and data augmentation techniques to medical benchmarks. Additionally, we believe that incorporating adversarial training for medical data can be a viable approach to enhance the robustness of the models.

## 5 Conclusion and Final Remark

**Conclusion.** In this study, we presented a comprehensive medical anomaly detection benchmark that encompasses six distinct benchmarks derived from five major medical domains. The benchmark integrated 15 SOTA AD algorithms, covering all major AD algorithm design paradigms. To ensure a thorough evaluation, we assessed the performance of these algorithms from multiple comparison perspectives, as detailed in the paper. This benchmark stands as the most extensive collection thus far, offering a comprehensive evaluation framework for medical AD algorithms.

**Limitation.** (1) As discussed in Sec. 3.1, since almost all data is collected in advanced countries, the data may have inherent geographical and sampling biases. (2) When evaluating the supported algorithms, we took great care to adhere to the hyper-parameter settings proposed in the original works. Thus not all hyper-parameters in our experiments achieved, or even close to, their optimal values for specific datasets. Our public codebase provides an interface to adjust hyper-parameters for each supporting algorithm. This feature empowers researchers to find the best settings that align with their research objectives. (3) Our evaluating in this Benchmark follows the one-for-one AD paradigm (one model for one subject/class). Recently, unified one-for-N models (one model for multiple classes) have shown many advantages. We believe that evaluating unified models on BMAD would provide insights for designing a more generic and versatile anomaly detection solution.

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

This is the supplementary document of our paper, entitled "BMAD: Benchmarks for Medical Anomaly Detection". It includes 4 sections, providing detailed information on datasets, supporting AD algorithms, experimental reproducibility, and evaluation metrics.

## A DATASETS IN BMAD

Our BMAD benchmark consists of six datasets sourced from five distinct medical domains, including brain MRI, retinal OCT, liver CT, chest X-ray, and digital histopathology. Due to the absence of specific anomaly detection datasets in the field of medical imaging, we construct these benchmark datasets by reorganizing and remixing existing medical image sets proposed for other purposes such as image classification and segmentation. Moreover, our codebase includes functionality for data reorganization, enabling users to generate new datasets tailored to their needs. In the this sections, we mainly focus on an overview of the original datasets and our data reorganization procedure.

### A.1 BRAIN MRI ANOMALY DETECTION AND LOCALIZATION BENCHMARK

The brain MRI anomaly detection benchmark is reorganized from the BraTS2021 dataset (Baid et al., 2021; Menze et al., 2014; Bakas et al., 2017).

#### A.1.1 BRATS2021 DATASET

The original BraTS2021 dataset is proposed for a multimodel brain tumor segmentation challenge. It provides 1,251 cases in the training set, 219 cases in validation set, 530 cases in testing set (non-public), all stored in NIFTI (.nii.gz) format. Each sample includes 3D volumes in four modalities: native (T1) and post-contrast T1-weighted (T1Gd), T2-weighted (T2), and T2 Fluid Attenuated Inversion Recovery (T2-FLAIR), accompanied by a 3D brain tumor segmentation annotation. The data size for each modality is 240 *240 *155.

Access and License: The BraTS2021 dataset can be accessed at http://braintumorsegmentation.org/. Registration for the challenge is required. As stated on the challenge webpage, "Challenge data may be used for all purposes, provided that the challenge is appropriately referenced using the citations given at the bottom of this page."

#### A.1.2 CONSTRUCTION OF BRAIN MRI AD BENCHMARK

After analyzing the BraTS2021 dataset, we built the brain MRI AD benchmark from the 3D FLAIR volumes. All data in our Brain MRI AD benchmark is derived from the 1,251 cases in the original training set. To account for variations in brain images at different depths, we specifically selected slices within the depth range of 60 to 100. Each extracted 2D slice was saved in PNG format and has an image size of 240 * 240 pixels. According to the tumer segmentation mask, we selected 7,500 normal samples to compose the AD training set, 3,715 samples containing both normal and anomaly samples (with a ratio of 1:1) for the test set, and a validation set with 83 samples that do not overlap with the test set. Fig. 4 illustrates the specific procedure we followed for data preparation, and Fig. 5 provides examples of our brain MRI AD benchmark.

### A.2 LIVER CT ANOMALY DETECTION AND LOCALIZATION BENCHMARK

We structure this benchmark from two distinct datasets, BTCV (Landman et al., 2015) and LiTS (Bilic et al., 2019). The anomaly-free BTCV set is taken to constitute the normal train set in this benchmark and CT scans in LiTs is exploited to form the evaluation and test data.

#### A.2.1 BTCV DATASET

BTCV (Landman et al., 2015) is introduced for multi-organ segmentation. It consists of 50 abdominal computed tomography (CT) scans taken from patients diagnosed with colorectal cancer and a retrospective ventral hernia. The original scans were acquired during the portal venous contrast phase and had variable volume sizes ranging from 512*512*85 to 512*512*198 and stored in nii.gz format.

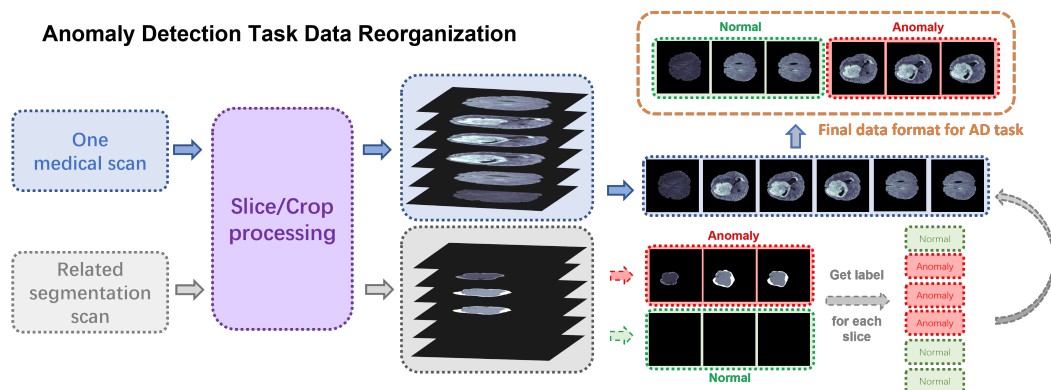

Figure 4: Diagram illustration of data preparation for the Brain MRI AD benchmark from 3D brain scans in BraTS2021.

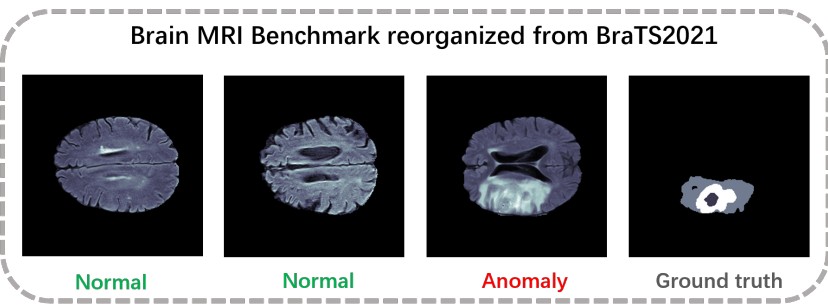

Figure 5: Visualization of our proposed Brain MRI benchmark.

Access and License: The original BTCV dataset can be accessed from 'RawData.zip' at https://www.synapse.org/#!Synapse:syn3193805/wiki/217753. Dataset posted on Synapse is subject to the Creative Commons Attribution 4.0 International (CC BY 4.0) license.

### A.2.2 LiTS Dataset

LiTS (Bilic et al., 2019) is proposed for liver tumor segmentation. It originally comprises 131 abdominal CT scans, accompanied by a ground truth label for the liver and liver tumors. The original LiTS is stored in the nii.gz format with a volume size of 512*512*432.

Access and License: LiTS can be downloaded from its Kaggle webpage at https://www.kaggle.com/datasets/andrewmvd/liver-tumor-segmentation. The use of the LiTS dataset is under Creative Commons Attribution-NonCommercial-ShareAlike(CC BY-NC-SA) (Bilic et al., 2023).

### A.2.3 Construction of Liver CT AD Benchmark

In constructing the liver CT AD benchmark, we made a decision not to include lesion-free regions from the LiTS dataset as part of the training set. This choice was based on our observation that the presence of liver lesions in LiTS leads to morphological changes in non-lesion regions, which could impact the performance of anomaly detection. Instead, we opted to use the lesion-free liver portion from the BTCV dataset to form the training set. The LiTS dataset, on the other hand, is reserved for testing the effectiveness of anomaly detection and localization.

For both datasets, Hounsfield-Unit (HU) of the 3D scans are transformed into grayscale with an abdominal window. The scans are then cropped into 2D axial slices, and the liver's Region of Interest is extracted based on the provided organ annotations. We perform slide intensity normalization with histogram equalization. To be more specific, for the construction of the normal training set in the liver CT AD benchmark, we utilized the provided segmentation labels in BTCV to extract the

liver region. From these scans, we extracted 2D slices of the liver with a size of 512 * 512, using the corresponding liver segmentation scans as a guide. The 2D slices were then converted to PNG format to serve as the final AD data. We selected 1542 slices to comprise the training set. To prepare the testing and validation sets, we sliced the data from LiTS and stored them in PNG format with dimensions of 512 * 512. Our testing and validation sets contain both healthy and abnormal samples. Fig. 6 demonstrates several samples in the Liver CT AD dataset. Fig. 6 provides visualization of the constructed Liver CT AD dataset.

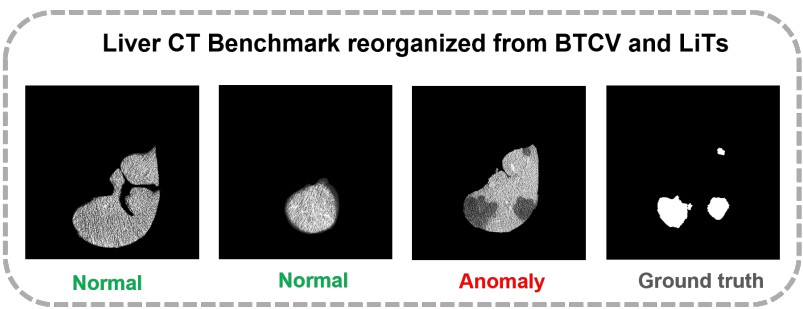

Figure 6: Visualization of our proposed Liver CT benchmark.

### A.3 RETINAL OCT ANOMALY DETECTION AND LOCALIZATION BENCHMARK

The BMAD datasets includes two different OCT anomaly detection datasets. The first one is derived from the RESC dataset (Hu et al., 2019) and support anomaly localization evaluation. The second is constructed from OCT2017 (Kermany et al., 2018), Which only support sample-level anomaly detection.

#### A.3.1 RESC DATASET

RESC (Retinal Edema Segmentation Challenge) dataset (Hu et al., 2019) specifically focuses on the detection and segmentation of retinal edema anomalies. It provides pixel-level segmentation labels, which indicate the regions affected by retinal edema. The RESC is provided in PNG format with a size of 512*1024 pixels.

Access and License: The original RESC dataset can be downloaded from the P-Net github page at https://github.com/CharlesKangZhou/P_Net_Anomaly_Detection. As indicated on the webpage, the dataset can be only used for the research community.

#### A.3.2 OCT2017 DATASET

OCT2017 (Kermany et al., 2018) is a large-scale dataset initially designed for classification tasks. It consists of retinal OCT images categorized into three types of anomalies: Choroidal Neovascularization (CNV), Diabetic Macular Edema (DME), and Drusen Deposits (DRUSEN). The images are continuous slices with a size of 512*496.

Access and License: OCT2017 can be downloaded at https://data.mendeley.com/datasets/rscbjbr9sj/2. Its usage is under a license of Creative Commons Attribution 4.0 International(CC BY 4.0).

#### A.3.3 PREPARATION OF OCT AD BENCHMARKS

To construct the OCT anomaly detection and localization dataset from RESC, we utilize the segmentation labels provided for each slice to get the label for AD setting. We select the normal samples from the original training dataset and adapt the original validation set into the AD setting for evaluation. The RESC is provided in PNG format with a size of 512*1024 pixels. On the other hand, on the OCT2017 dataset, we specifically select the disease-free samples from the original training set as our training data for the anomaly detection task. The test set is further divided into evaluation data and testing data for AD setting. Fig. 7 demonstrates several examples in the two OCT AD datasets.

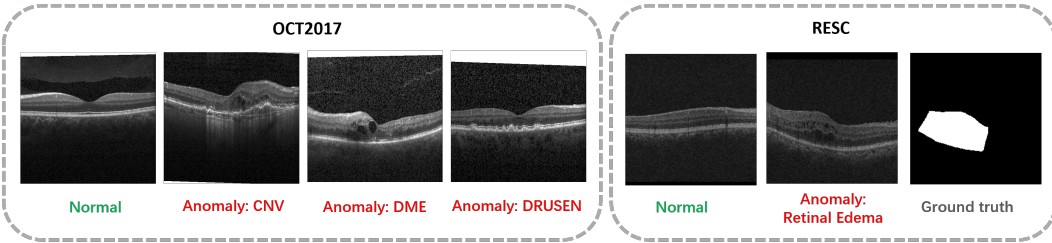

Two Retinal OCT Benchmarks reorganized from OCT2017 and RESC

Figure 7: The Retinal OCT benchmarks consist of two separate datasets, each representing different anomaly types. These datasets are used to evaluate and benchmark various methods in the field of retinal OCT imaging. The datasets are designed to assess the performance of algorithms in detecting and localizing specific anomalies in retinal images.

### A.4 CHEST X-RAY ANOMALY DETECTION BENCHMARK

#### A.4.1 RSNA DATASET

RSNA (Wang et al., 2017), short for RSNA Pneumonia Detection Challenge, is originally provided for a lung pneumonia detection task. The 26,684 lung images are associated with three labels: "Normal" indicates a normal lung condition, "Lung Opacity" indicates the presence of pneumonia, "No Lung Opacity/Not Normal" represents a third category where some images are determined to not have pneumonia, but there may still be some other type of abnormality present in the image. All images in RSNA are in DICOM format.

**Access and License:** RSNA can be accessed by https://www.kaggle.com/competitions/rsna-pneumonia-detection-challenge/overview. Stated in the section of Competition data: A. Data Access and Usage, "... you may access and use the Competition Data for the purposes of the Competition, participation on Kaggle Website forums, academic research and education, and other non-commercial purposes."

#### A.4.2 PREPARATION OF CHEST X-RAY AD BENCHMARK

We utilized the provided image labels for data re-partition. Specifically, "Lung Opacity" and "No Lung Opacity/Not Normal" were classified as abnormal data. The reorganized AD dataset including 8000 normal images as training data, 1490 images with 1:1 normal-versus-abnormal ratio in the validate set, and 17194 images in the test set. Examples of the chest X-ray dataset are provided in Fig. 8.

### A.5 DIGITAL HISTOPATHOLOGY ANOMALY DETECTION BENCHMARK

#### A.5.1 CAMELYON16 DATASET

The Camelyon16 dataset (Bejnordi et al., 2017) was initially utilized in the Camelyon16 Grand Challenge to detect and classify metastatic breast cancer in lymph node tissue. It comprises 400 whole-slide images (WSIs) of lymph node sections stained with hematoxylin and eosin (H&E) from breast cancer patients. Among these WSIs, 159 of them exhibit tumor metastases, which have been annotated by pathologists. The WSIs are stored in standard TIFF files, which include multiple down-sampled versions of the original image. In Camelyon16, the highest resolution available is on level 0, corresponding to a magnification of 40X.

Access and Licence: The original Camelyon16 dataset can be found at https://camelyon17.grand-challenge.org/Data/. It is under a license of Creative Commons Zero 1.0 Universal Public Domain Dedication(CC0).

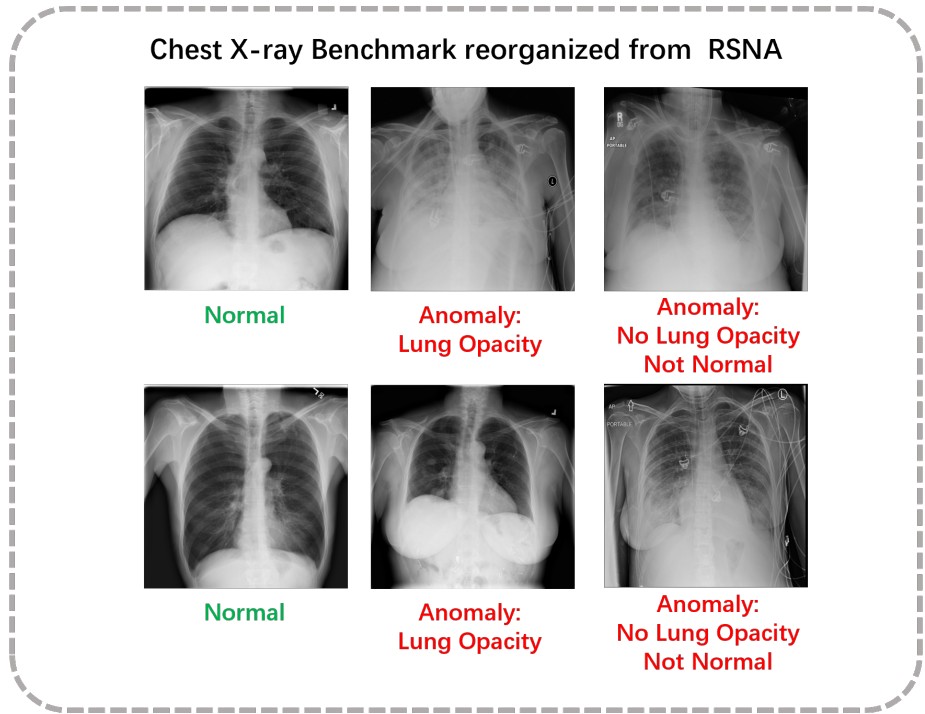

Figure 8: Our proposed chest X-ray benchmark consists two types of anomalies. These anomalies are clearly labeled in the images, and all of them are considered as anomaly samples.

### A.5.2 PREPARATION OF HISTOPATHOLOGY AD BENCHMARK

To ensure a comprehensive evaluation of anomaly detection models for histopathology images, considering their unique characteristics such as large size, we opted to assess AD models at the patch level. To construct the benchmark dataset, we randomly extracted 5,088 normal patches from the original training set of Camelyon16, which consisted of 160 normal WSIs. These patches were utilized as training samples. For the validation set, we cropped 100 normal and 100 abnormal patches from the 13 testing WSIs. Likewise, for the testing set, we extracted 1,000 normal and 1,000 abnormal patches from the 115 testing WSIs in the original Camelyon16 dataset. Each cropped patch was saved as a PNG image with dimensions of 256 * 256 pixels. Fig. 9 presents several examples in the constructed histopathology AD benchmark.

### A.6 REMARKS ON BENCHMARK DATASETS

We observed that there were biases present in the original datasets, which may impact the performance of the models. For instance, in the chest dataset, the results may be influenced by the uneven gender distribution. Additionally, in the liver CT benchmark, the performance can be affected by the bias introduced by the cropped area.

## B SUPPORTED AD MODELS

Fig. 10 provides conceptual illustration of various AD architectures from the feature embedding-based methods and data reconstruction-based approaches. The specific experimental settings for each of the supported methods are specified as follows.

**PaDiM (Defard et al., 2021)** leverages a pre-trained convolutional neural network (CNN) for its operations and does not require additional training. In our experiments, we separately evaluated all benchmarks using two backbone networks: ResNet-18 and WideResnet-50. For the dimension reduction step, we retained the default number of features as specified in the original setting. Specifically, we used 100 features for ResNet-18 and 550 features for WideResnet-50. These default values

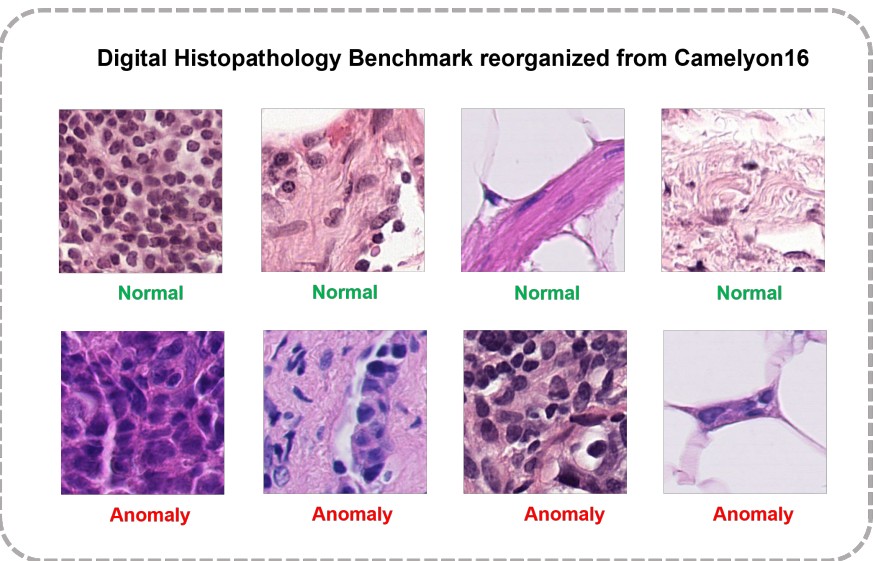

Figure 9: Examples of the digital histopathology AD benchmark. Unlike other medical image AD benchmarks, histopathology images shows higher diversities in tissue components.

were chosen based on the original implementation and can serve as a starting point for further experimentation and fine-tuning if desired.

**STFPM (Yamada & Hotta, 2021)** utilized feature extraction from a Teacher-student structure. In our experiments, we evaluated all benchmarks separately using two backbone networks: ResNet-18 and WideResnet-50. We employed a SGD optimizer with a learning rate of 0.4. Additionally, we followed the original setting with a parameter with a momentum of of 0.9 and weight decay of 1e-4 for SGD. These settings were chosen based on the original implementation and can be adjusted for further experimentation if desired.

**Patchcore (Roth et al., 2022)** is a memory-based method that utilizes coreset sampling and neighbor selection. In our experiments, we evaluated Patchcore using two backbone networks: ResNet-18 and WideResnet-50. We followed the default hyper-parameters of 0.1 for the coreset sampling ratio and 9 for the chosen neighbor number. These values were chosen based on the original implementation.

**RD4AD (Deng & Li, 2022b)** utilizes a wide ResNet-50 as the backbone network and applies the Adam optimizer with a learning rate of 0.005. In addition, we follow the defeat set of the beta1 and beta2 parameters to 0.5 and 0.99, respectively. For the anomaly score of each inference sample, the maximum value of the anomaly map is used. These settings were determined based on the original implementation of RD4AD and can be adjusted if needed.

**DRAEM (Zavrtanik et al., 2021a)** is a anomaly augmentation reconstruction-based method utilized U-Net structure. The learning rate used for two sub network training is 1e-4, and the Adam optimizer is employed. For the remaining settings, we follow the default configurations specified in the original work.

**CFLOW (Gudovskiy et al., 2022)** is a normalizing flows-based method. We utilized WideResnet-50 as backbone and Adam optimizer with a learning rate of 1e-4 for all benchmarks' experiments. And we follow the original parameter settings, including the selection of 128 for the number of condition vectors and 1.9 as clamp alpha value.

**CFA (Lee et al., 2022)** is also a memory bank-based algorithm. We employs a WideResnet-50 backbone and follows the parameter settings outlined in the original paper. The method utilizes 3 nearest neighbors and 3 hard negative features. A radius of 1e-5 is utilized for searching the soft boundary within the hypersphere. The model is trained using the Adam optimizer with a learning rate of 1e-3 and a weight decay of 5e-4. These specific parameter configurations play a crucial role in achieving the desired performance and effectiveness of the CFA approach, as determined by the original research paper or implementation.

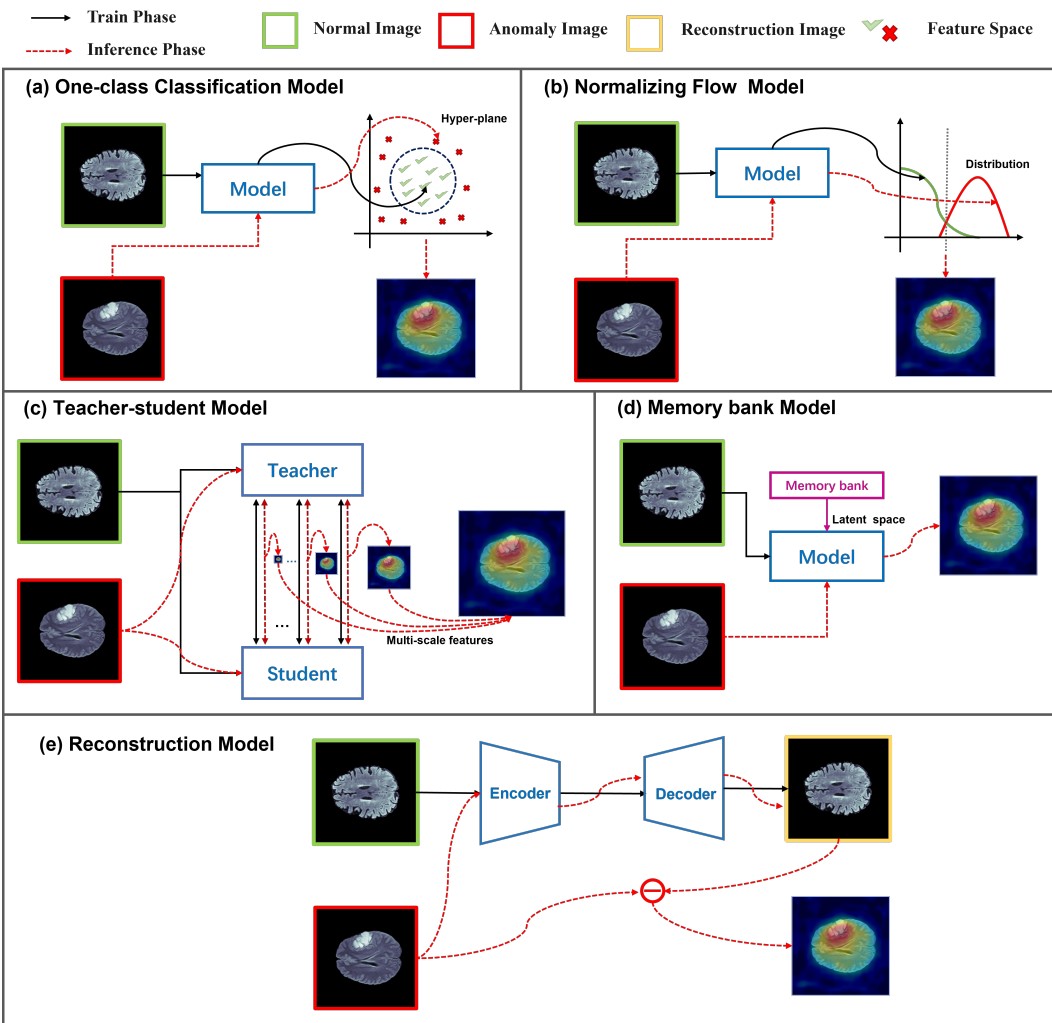

Figure 10: Conceptual illustration of various AD models. The one-class classification model, normalizing flow model, teaching-student model and memory bank model detects anomalies in the embedding space, and the reconstruction based method takes a generative model as its backbone for pixel-level anomaly comparison between the original query and reconstruction.

**MKD (Salehi et al., 2021)** utilizes the VGG16 backbone for feature extraction, and only the parameters of the cloner are trained. We follow the defeat setting with a batch size of 64. The learning rate is set to 1e-3 using the Adam optimizer. Additionally, the $\lambda$ value is set to 1e-2, which represents the initial amount of error assigned to each term on the untrained network. These parameter settings are have been chosen based on the original research paper.

**UTRAD (Chen et al., 2022)** is based on Transformer backbone with a ReLu activation function. We trained the model with a defeat parameters setting: batch size of 8 and an Adam optimizer with a learning rate of 1e-4. The parameter settings are have been chosen based on the original research paper.

**CutPaste (Li et al., 2021a)** utilizes a Resnet-18 backbone. The backbone is frozen for the first 20 epochs of training. We trained the model using an SGD optimizer with a learning rate of 0.03. And the batch size for training is following to the defeat parameter, set to 64.

**GANomaly (Akcay et al., 2018)** is trained using an Adam optimizer with a learning rate of 2e-4. The $\beta1$ and $\beta2$ parameters of the Adam optimizer are set to 0.5 and 0.999, respectively, following the original work. The weights assigned to different loss components are also set according to the

original setting: a weight of 1 for the adversarial loss, a weight of 50 for the image regeneration loss, and a weight of 1 for the latent vector encoder loss. These parameter values have been chosen based on the original research paper and are crucial for the performance and effectiveness.

**DeepSVDD (Ruff et al., 2018)** utilizes a LeNet as its backbone and is trained using an Adam optimizer with a learning rate of 1e-4. The model training follows the setting of weight decay as 0.5e-7 and a batch size of 200. These parameter values have been chosen based on the original research paper or implementation.

**f-AnoGAN (Schlegl et al., 2019)** is a generative network that requires two-stage training. During the training process, we use an Adam optimizer with a batch size of 32 and a learning rate of 2e-4. Additionally, the dimensionality of the latent space is set to 100. These parameter settings have been chosen based on the original research paper.

**CS-Flow (Rudolph et al., 2022)** is trained using specific hyper-parameter settings. During the flow process, a clamping parameter of 3 is utilized to restrict the values. Gradients are clamped to a value of 1 during training. The network is trained with an initial learning rate of 2e-4 using the Adam optimizer, and a weight decay of 1e-5 is applied. These hyper-parameter settings have been determined through a process of optimization and are considered optimal for the CS-Flow method.

**SimpleNet (Liu et al., 2023)** was trained using the original hyper-parameters and includes two main modules. We retained the original parameters for the adapter and the Gaussian noise generation module. The results are based on the best performance achieved on the validation set during the top 40 training epochs, following the original settings.

## C  EXPERIMENT REPRODUCIBILITY

We conducted benchmarking using the Anomalib (Akcay et al., 2022) for CFA, CFlow, DRAEM, GANomaly, PADIM, PatchCore, RD4AD, and STFPM. For the remaining algorithms, we provided a comprehensive codebase for training and inference with all proposed evaluation metrics functions. By utilizing these codebases and following the instructions provided, researchers can replicate and reproduce our experiments effectively. In addition to the codebase, we also provide pre-trained checkpoints for different benchmark on our webpage.

## D  MATHEMATICAL METRICS

### D.1  AUROC

AUROC refers to the area under the ROC curve. It provides a quantitative value showing a trade-off between True Positive Rate (TPR) and False Positive Rate (FPR) across different decision thresholds.

$$AUROC = \int_0^1 (TPR)\mathrm{d}(FPR) \tag{1}$$

- To calculate the pixel-level AUROC, different thresholds are applied to the anomaly map. If a pixel has an anomaly score greater than the threshold, the pixel is anomalous. Over an entire image, the corresponding TPR and FPR pairs are recorded for a ROC curve and the area under the curve is calculated as the final metric.
- To calculate the image-level AUROC, each model independently calculates an anomaly score from the anomaly map as a sample-level evaluation metric. Then different thresholds are applied to determine if the sample is normal or abnormal. Then the corresponding TPR and FPR pairs are recorded for estimating the ROC curve and sample-level AUROC value.

### D.2  PER-REGION OVERLAP (PRO)

We utilized PRO, a region-level metric, to assess the performance of fine-grained anomaly detection. To compute PRO, the ground truth is decomposed into individual unconnected components. Let $A$ denote the set of pixels predicted to be anomalous. For connected components $k$, $C_k$ represents the

set of pixels identified as anomalous. PRO can then be calculated as follows,

$$PRO = \frac{1}{N} \sum_k \frac{|A \cap C_k|}{|C_k|},$$ (2)

where $N$ represents the total number of ground truth components in the test dataset.

## D.3 DICE SCORE

The Dice score is an important metric in medical image segmentation, evaluating the similarity between segmented results and reference standards. It measures the pixel-level overlap between predicted and reference regions, ranging from 0 (no agreement) to 1 (perfect agreement). Higher Dice scores indicate better segmentation consistency and accuracy, making it a commonly used metric in medical imaging for comparing segmentation algorithms.

It should be noted that the Dice score is a threshold dependent metric. It requires different threshold values for different models and datasets to better suit the specific task. Therefore, we opted to not include the DICE comparison in the main experimentation.

[Remark:] Due to the significance of DICE in medical segmentation, our codebase also includes a Dice function for its potential usage. For reference, Table 3 provides the Dice scores for the suppoeted AD methods with the threshold 0.5. By adjusting the threshold for each result, it is possible to achieve higher performance.

| Benchmarks | BraTS2021 | BTCV + LiTs | RESC |
|---|---|---|---|
| DRAEM (Zavrtanik et al., 2021a) | $19.31 \pm 5.52$ | $9.38 \pm 0.78$ | $33.51 \pm 3.52$ |
| UTRAD (Chen et al., 2022) | $7.27 \pm 0.06$ | $2.33 \pm 0.06$ | $22.81 \pm 0.36$ |
| MKD (Salehi et al., 2021) | $28.89 \pm 0.72$ | $\underline{14.92 \pm 0.23}$ | $43.53 \pm 1.10$ |
| RD4AD (Deng & Li, 2022b) | $28.28 \pm 0.48$ | $10.72 \pm 2.50$ | $33.51 \pm 3.52$ |
| STFPM (Yamada & Hotta, 2021) | $25.40 \pm 0.82$ | $8.87 \pm 2.52$ | $49.23 \pm 0.23$ |
| PaDiM (Defard et al., 2021) | $25.84 \pm 1.20$ | $4.50 \pm 0.46$ | $38.30 \pm 0.89$ |
| PatchCore (Roth et al., 2022) | $\underline{32.82 \pm 0.59}$ | $10.49 \pm 0.23$ | $\underline{57.04 \pm 0.21}$ |
| CFA (Lee et al., 2022) | $30.22 \pm 0.32$ | $\underline{14.93 \pm 0.08}$ | $36.57 \pm 0.18$ |
| CFLOW (Gudovskiy et al., 2022) | $19.50 \pm 2.73$ | $7.58 \pm 3.16$ | $44.83 \pm 1.78$ |
| SimpleNet (Liu et al., 2023) | $28.96 \pm 1.73$ | $12.26 \pm 2.41$ | $30.28 \pm 1.64$ |

Table 3: AD DICE comparison over the benchmarks in BMAD. The top method for each metric are underlined. Note that Dice is a threshold-dependent metric. The results in the table is obtained with threshold of 0.5. By adjusting the threshold for each result, it is possible to achieve higher performance.

