# OpenReview forum: "BMAD: Benchmarks for Medical Anomaly Detection"
_ICLR.cc/2024/Conference — Submitted to ICLR 2024_

### Official Review · Reviewer_csZB · 2023-10-31

**Soundness:** 4 excellent
**Presentation:** 3 good
**Contribution:** 2 fair
**Rating:** 3
**Confidence:** 4

**Summary:**

This paper introduces BMAD (Benchmarks for Medical Anomaly Detection), a benchmark dataset for anomaly detection (AD) containing six reorganized public datasets from five medical domains, three evaluation metrics, and fifteen AD algorithms. Analysis of the relative advantages and disadvantages of the AD algorithms is included. A stated motivation is the lack of existing uniform, comprehensive, standardized and fair benchmarks for medical anomaly detection.

**Strengths:**

-	Contributes a well-organized benchmark dataset to the community
-	Careful evaluation and analysis of implemented algorithms including hyperparameter optimization and multiple executions

**Weaknesses:**

-	Limited technical novelty
-	Definition and scope of anomalies possibly more meaningful if further extended, especially in the medical domain

**Questions:**

1. The overarching claim of a “universal and fair” benchmark appears difficult to substantiate, as representation from five domains necessarily remains a small subset of possible image anomaly cases in medicine. The degree of fairness regarding domain selection might be further justified. For example, thin structures appear relatively less represented.
2. As raised in the Introduction, a common practical concern for anomaly detection in the medical domain is when some (more-common) disease classes are known and labelled/annotated, but other rarer classes not in the initially known set of diseases may occur. Some analysis of such tasks might thus be relevant.
3. A number of minor spelling/grammatical errors might be addressed, e.g. “a fair comparing among these methods”, “well-structure, easy-use codebase”, “samll tumor localization”, “large pre-train models”, etc.

---

> ### Author Response · Authors · 2023-11-23
> **Response to Reviewer csZB**
>
> Thank you for acknowledging our effort in curating this benchmark. Our point-to-point feedback is as follow.
>
> **Regarding technical novelty**
>
> We appreciate your concern, as it gives us an opportunity to clarify the focus of our work. This work primarily revolves around providing a well-structured benchmark for medical anomaly detection, rather than introducing new algorithms. However, the contributions of our work extend beyond data curation. Notably, the field of medical anomaly detection has suffered from a lack of a standardized evaluation benchmark, resulting in data citation errors and inconsistent experimental settings in numerous published studies. Our benchmark aims to bridge this critical gap by offering a standardized and unified codebase and evaluation protocol for fair algorithm assessment. Both the complete codebase and a comprehensive analysis of existing algorithms across various medical modalities facilitate the development of improved anomaly detection models for medical data.
>
> **Regarding "universal and fair" benchmark**
>
> In our study, we use the term "universal and fair" to emphasize that the codebase and evaluation protocol remain consistent across all evaluated algorithms and datasets. This approach is intended to rectify the prevailing issues of inconsistent evaluation practices and data partitioning in the current literature. We apologize if the use of "universal and fair" has caused any confusion for the reviewer.
>
> **Regarding the purpose of medical anomaly detection**
>
> it's important to note that the primary goal of anomaly detection differs from classifying known diseases; it is focused on detecting abnormalities in an open-set setting. In the context of medical image classification, assembling a comprehensive training set covering all possible scenarios, especially rare diseases and unprecedented anomalies, is impractical. The inherent open-set nature of medical data collection implies that supervised classification based on an incomplete training set may lead to misclassification. Anomaly detection serves as a solution to address this open-set problem and complements the limitations of supervised learning. We plan to incorporate this discussion in our revision.
>
> **Regarding typos**
>
> We would like to thank the reviewer for pointing out these typos. We will thoroughly proofread the manuscript to address these issues.

---

### Official Review · Reviewer_jY9s · 2023-10-31

**Soundness:** 2 fair
**Presentation:** 2 fair
**Contribution:** 1 poor
**Rating:** 3
**Confidence:** 4

**Summary:**

The paper proposes a benchmark for anomaly detection in medicine. The paper presents six medical datasets as an evaluation benchmark for anomaly detection. The paper evaluates multiple state-of-the-art algorithms on these datasets using the evaluation metrics of AUROC, Per-Region Overlap, and Dice. The benchmark is integrated in a code base and aims to facilitate and simplify the use of the data and algorithms in the medical image analysis community. The advantages and disadvantages of the evaluated algorithms are evaluated and discussed.

**Strengths:**

- the paper proposes a standardized access to multiple medical image datasets
- standardized implementation of multiple algorithms
- paper is well written and easy to follow

**Weaknesses:**

I appreciate the efforts put into this submission, but I find it challenging to pinpoint any technical, methodological, or experimental contribution that aligns with the standards of acceptance for ICLR. For example, the proposed datasets have been studied in light of anomaly detection before. Overall, I think the initiative can be very useful for the medical image analysis community, and I would therefore encourage resubmission to a more applied medical image analysis venue.

**Questions:**

-

---

> ### Author Response · Authors · 2023-11-23
> **Response to Reviewer jY9s**
>
> We appreciate your valuable feedback on our paper. Below is our point-to-point response.
>
> **Regarding technical contribution**
>
> We would like to thank the reviewer for acknowledging our effort in curating this benchmark. Rather than introducing new algorithms, this work primarily revolves around providing a well-structured benchmark (e.g. unified datasets and evaluation protocols) for medical anomaly detection, rather than introducing new algorithms. However, the contributions of our work extend beyond data curation. Notably, the field of medical anomaly detection has suffered from a lack of a standardized evaluation benchmark, resulting in data citation errors and inconsistent experimental settings in numerous published studies. Our benchmark aims to bridge this critical gap by offering a standardized and unified codebase and evaluation protocol for fair algorithm assessment. Both the complete codebase and a comprehensive analysis of existing algorithms across various medical modalities facilitate the development of improved anomaly detection models for medical data.
>
> **Regarding submission venue**
>
> We would like to thank the review for bringing this concern. Aside from the work itself, here is the the second motivation that we submitted this benchmark to ICLR. Almost all SOTA anomaly detection methods have been tested on natural images or industry images. However, due to the significant domain gap between medical images and natural/industrial images, these methods do not perform well in medical anomaly detection (as shown in the manuscript). Since anomaly detection itself is a form of data-efficient learning, we hope that the results and insights from this benchmark can attract more scholars to focus on this field and propose more generic methods to address these issues.

---

### Official Review · Reviewer_Y82y · 2023-11-02

**Soundness:** 2 fair
**Presentation:** 2 fair
**Contribution:** 2 fair
**Rating:** 5
**Confidence:** 4

**Summary:**

Anomaly detection is crucial in fields like medical imaging, but there's a lack of standardized evaluation benchmarks. To address this, a comprehensive benchmark for assessing anomaly detection methods in medical images was introduced in this paper. This benchmark includes datasets from five medical domains, 15 state-of-the-art algorithms, and provides a robust framework for advancing anomaly detection in medical imaging.

**Strengths:**

The paper addresses the lack of a universal and fair benchmark for evaluating anomaly detection (AD) methods on medical images. This is good contribution to the field, as it fills a significant gap and provides a standardized platform for evaluating AD methods in the context of medical imaging.

The paper demonstrates a good level of quality in terms of dataset curation and algorithm integration. It organizes six datasets from five different medical domains, ensuring diversity and representation of real-world medical scenarios.

The paper is clear in its presentation. It is well-structured and provides detailed information about the benchmark, including the datasets, evaluation metrics, and algorithms used.

The benchmark is very importance for the field of anomaly detection in medical imaging. Medical imaging have a crucial role in diagnosing / monitoring various diseases. A standard benchmark can lead to the development of more reliable and robust AD methods, ultimately benefiting healthcare and patient outcomes.

The presence of Table 3, which provides insights into both the inference times and the performance of the AD models, is interesting as it not only facilitates a more robust comparison of the models but also helps researchers when assessing the efficiency of their proposed methodologies in the context of anomaly detection.

**Weaknesses:**

Regarding table 2, it would be highly recommended to represent the results in graphs for each data set (showing the relationship between methodology and performance). This would allow a more clear visualization and facilitate the comparison of the results between the different methodologies, improving understanding and the capacity to identify trends and variations.

While the paper mentions a well-structured codebase, it's crucial to provide insights into how this codebase is organized and made available to the community. Detailed documentation and code accessibility are essential for other researchers to replicate and extend the experiments.

The paper acknowledges that most of the data used in the benchmark is collected in advanced countries, which may introduce geographical and sampling biases. This limitation could potentially affect the generalizability of the benchmark to a broader range of medical imaging scenarios. To address this, the paper could suggest ways to mitigate these biases.

The paper mentions that the hyperparameter settings for the evaluated algorithms were based on the original works, and not all hyperparameters achieved their optimal values for specific datasets. This could potentially lead to suboptimal performance for some algorithms. To address this, the paper could provide recommendations or guidelines for tuning hyperparameters to improve the performance of the AD algorithms on the benchmark datasets. This would make the benchmark more valuable.

In addition to the mentioned quantitative metrics, incorporating qualitative analysis and insights from medical professionals regarding the benchmark's practical applicability in real-world anomaly detection scenarios would enhance its credibility and utility.

**Questions:**

As more algorithms and datasets become available, how will the benchmark be updated and maintained?

What are the plans for including new datasets and algorithms in the future?

---

> ### Author Response · Authors · 2023-11-22
> **Response to Reviewer Y82y**
>
> We would like to thank the reviewer for recognizing our work and the constructive comments. Below are our point-to-point response to your comments
>
> **Regarding evaluation visualizations:**
>
> Thanks for this comment. We plan to add bar plots of the numerical results in Table 2 and more anomaly detection examples in the revision.
>
> **Regarding codebase availability:**
>
> The codebase and the curated datasets will be posted on a dedicated GitHub page for public access. Detailed documentation will be provided there as well. We plan to maintain the GitHub page to include more datasets and new algorithms.
>
> **Regarding bias in datasets:**
>
> As commented in our manuscript, we realized the bias in the current datasets. We plan to include more medical anomaly detection datasets in this benchmark. We will pay attention to this issue in data preparation and hopefully the new datasets will help address the bias issue.
>
> **Regarding hyperparameter setting:**
>
> In this study, we did not extensively fine-tune hyper-parameters. Instead, we experimented with various combinations to achieve model convergence when the original settings don’t work. We kept hyper-parameters related to loss terms consistent with the original papers. Detailed experimental data and trials will be included in the revision.

---

### Official Review · Reviewer_SMmX · 2023-11-06

**Soundness:** 3 good
**Presentation:** 3 good
**Contribution:** 3 good
**Rating:** 6
**Confidence:** 3

**Summary:**

This paper proposes a benchmark study for medical anomaly detection including 1) reorganizing 6 existing medical datasets and 2) running 15 SOTA algorithms evaluated on 3 metrics.

**Strengths:**

1. The authors utilize proper preprocessing techniques to reorganize the six datasets and systematically run baselines with reasonable evaluation metrics.

**Weaknesses:**

1. As also noted by the authors, ethical and fairness concerns exist among the datasets and need to be addressed properly.

2. Some part of the writing needs to be improved (e.g. "defeat->default parameters setting").

**Questions:**

1. It would be more helpful to discuss the impact of hyper-parameters especially for methods that the authors find hard to achieve optimal values reported in the original paper.

2. From the section 4.1 implementation details, the authors seem to follow default settings first and then tune the learning rate/threshold only. Other important hyper-parameters, such as the weighting terms in losses, seem to be missing. It would also be very helpful to document the results using each combination so readers would know what works and what doesn't, which will also improve the reproducibility of a benchmark study.

3. It would be interesting to have a section discussing the robustness of each algorithm analyzed in training (e.g. which is easier to converge).

---

> ### Author Response · Authors · 2023-11-22
> **Response to Reviewer SMmX**
>
> We appreciate your valuable feedback on our paper, particularly your insights on hyper-parameters and model training efficiency.
>
> **Regarding hyper-parameter settings:**
>
> In our benchmark study of anomaly detection algorithms for medical images, we did not extensively fine-tune hyper-parameters. Instead, we experimented with various combinations to achieve convergence when the original settings were insufficient. We kept hyper-parameters related to loss terms consistent with the original papers. Detailed experimental data and trials will be included in the revision.
>
> **Regarding model training efficiency:**
>
> We acknowledge your suggestion to include a section on the robustness of each algorithm during training. This will provide valuable insights into their behavior under different conditions. We will thoroughly address this aspect in the revised manuscript.

---

### Meta-Review · Area_Chair_J9J9 · 2023-12-11

**Metareview:**

The paper introduces a benchmark for anomaly detection in medical images, named BMAD. This benchmark includes six reorganized public datasets from various medical domains, evaluates fifteen state-of-the-art anomaly detection algorithms using three metrics. The paper aims to address the lack of standardized benchmarks for evaluating anomaly detection methods in medical imaging. However, the paper suffers from issues including its limited technical novelty, as the introduced benchmark dataset doesn't substantially advance the methodology of anomaly detection. The definition and scope of anomalies should be more meaningful, especially in the medical domain. The claim of a "universal and fair" benchmark might be challenging to substantiate, considering representation from five medical domains may not cover the full spectrum of possible image anomaly cases in medicine. The paper also lacks further justification of the fairness regarding domain selection. Additionally, the practical concern of anomaly detection in scenarios where some diseases are known and labeled, while others are unknown, could be explored for added relevance. The meta-reviewer concurs that the paper's fundamental scientific contribution is unclear, and, on the whole, the drawbacks of the paper outweigh its merits.

**Justification For Why Not Higher Score:**

While the paper offers a well-organized benchmark dataset and carries out a comprehensive evaluation of existing algorithms, it lacks significant technical novelty. Additionally, there is a critical need for a more extensive exploration of the definition and scope of anomalies, particularly within the medical domain.

**Justification For Why Not Lower Score:**

N/A

---

### Decision · Program_Chairs · 2024-01-16

Reject